LHCHWG-2025-001
IPPP/25/01, TUM-HEP-1549/25, UWThPh 2024-25,
TTK-25-01, P3H-25-001, TIF-UNIMI-2025-3,
MS-TP-25-02, PSI-PR-25-03, ZU-TH 08/25

# State-of-the-art cross sections for $t\bar{t}H$: NNLO predictions matched with NNLL resummation and EW corrections

**Roger Balsach**[1] **Alessandro Broggio**[2] **Simone Devoto**[3] **Andrea Ferroglia**[4] **Rikkert Frederix**[5] **Massimiliano Grazzini**[6] **Stefan Kallweit**[6] **Anna Kulesza**[1] **Javier Mazzitelli**[7] **Leszek Motyka**[8] **Davide Pagani**[9] **Benjamin D. Pecjak**[10] **Chiara Savoini**[11] **Tomasz Stebel**[8] **Malgorzata Worek**[12] **Marco Zaro**[13]

[1] *Institute for Theoretical Physics, University of Münster, D-48149 Münster, Germany*

[2] *Faculty of Physics, University of Vienna, Boltzmanngasse 5, A-1090 Vienna, Austria*

[3] *Department of Physics and Astronomy, Ghent University, 9000 Ghent, Belgium*

[4] *Physics Department, New York City College of Technology, The City University of New York, 300 Jay Street, Brooklyn, NY 11201, USA & The Graduate School and University Center, The City University of New York, 365 Fifth Avenue, New York, NY 10016, USA*

[5] *Department of Physics, Lund University, SE-223 63 Lund, Sweden*

[6] *Physik Institut, Universität Zürich, 8057 Zürich, Switzerland*

[7] *Paul Scherrer Institut, 5232 Villigen PSI, Switzerland*

[8] *Institute of Theoretical Physics, Jagellonian University, S.Łojasiewicza 11, 30-348 Kraków, Poland*

[9] *INFN, Sezione di Bologna, Via Irnerio 46, 40126 Bologna, Italy*

[10] *Institute for Particle Physics Phenomenology, Department of Physics Durham University, Durham DH1 3LE, United Kingdom*

[11] *Technical University of Munich, TUM School of Natural Sciences, Physics Department, James-Franck-Straße 1, 85748 Garching, Germany*

[12] *Institute for Theoretical Particle Physics and Cosmology, RWTH Aachen University, D-52056 Aachen, Germany*

[13] *TIFLab, Università degli Studi di Milano & INFN, Sezione di Milano, Via Celoria 16, 20133 Milano, Italy*

*E-mail:* rbalsach@uni-muenster.de, alessandro.broggio@univie.ac.at, simone.devoto@ugent.be, aferroglia@citytech.cuny.edu, rikkert.frederix@fysik.lu.se, grazzini@physik.uzh.ch, stefan.kallweit@physik.uzh.ch, anna.kulesza@uni-muenster.de, javier.mazzitelli@psi.ch, leszekm@th.if.uj.edu.pl, davide.pagani@bo.infn.it, chiara.savoini@tum.de, tomasz.stebel@uj.edu.pl, worek@physik.rwth-aachen.de, marco.zaro@mi.infn.it

ABSTRACT: We present new, state-of-the-art predictions for the associated production of the SM Higgs boson with top quarks, computed in accordance with the recommendations of the LHC Higgs Working Group. The NNLO QCD predictions, derived through suitable approximations of the two-loop virtual contribution, are supplemented with soft-gluon resummation up to NNLL accuracy. Two distinct resummation frameworks are employed – one based on direct QCD and the other on soft-collinear effective theory – and their features are compared in detail. These results are further combined with the complete-NLO corrections, yielding the most precise SM predictions for this process to date. The relevant sources of theoretical uncertainties are thoroughly estimated and discussed.

## Contents

## 1 Introduction

After the discovery of the Higgs boson at the Large Hadron Collider (LHC) at CERN [1, 2], an unprecedented campaign of measurements has begun. Such a campaign aims to thoroughly scrutinise the properties of the newly discovered particle, on the one hand to verify the consistency of the Standard Model (SM) of fundamental interactions, and on the other to exploit this particle as a magnifying glass for new, yet undiscovered physics phenomena. More than a decade after its discovery and on the cusp of the LHC Run III, the SM description of the Higgs sector has been corroborated at the level of a few percent. Indeed, the new combination of all the Run II data yields $\mu = 1.002 \pm 0.036\,(\text{th.}) \pm 0.029\,(\text{stat.}) \pm 0.033\,(\text{syst.}) = 1.002 \pm 0.057$ [3, 4], where $\mu$ is a common signal-strength parameter that quantifies the agreement between the observed signal yields from all production modes and decay channels, and the corresponding SM expectations. The uncertainties associated with the new measurement reflect a 4.5-fold improvement in precision compared to that achieved at the time of discovery. Currently, the theoretical uncertainties in both signal

and background modelling as well as the experimental statistical and systematic uncertainties contribute at a similar level. Achieving this level of accuracy, as well as matching that of the Run III data, requires continuously improving theoretical predictions for the various Higgs boson production processes and their backgrounds.

Among the studied properties of the Higgs boson, its coupling to fermions (quark and leptons) has been among the most recently established. Experimental evidence for the Higgs boson interaction with third-generation fermions (top and bottom quarks, and the $\tau$ lepton) was obtained only six years after the Higgs discovery [5–13], while evidence for its coupling to second-generation fermions followed in 2020, with the observation of the Higgs boson interaction with muons [14]. More recently, tighter constraints have been placed on the charm-quark Yukawa coupling by the ATLAS and CMS collaborations [15, 16]. The importance of probing the Higgs couplings to fermions stems from the fact that such couplings are mediated by the so-called Yukawa interaction, a novel type of interaction that has never been observed among elementary particles before. Furthermore, since the known fermions acquire their masses via the same interaction, the Higgs-fermion interactions can shed some light on the observed pattern of quark and lepton masses in the SM.

Of all fermions, the top quark is the heaviest and, consequently, the most strongly coupled to the Higgs boson. The top-quark Yukawa coupling, $y_t$, can be probed indirectly via a number of processes: the Higgs gluon-fusion, where the top quark appears in a closed loop, the four top production $pp \rightarrow t\bar{t}t\bar{t}$, which is sensitive to $y_t$ at the tree level via diagrams featuring an off-shell Higgs propagator [17–19], and also the top-pair production $pp \rightarrow t\bar{t}$, where high-precision measurements allow sensitivity to effects induced by $y_t$ via EW loops [20–24]. In all these cases, the extraction of $y_t$ depends on the assumptions made about new-physics effects, particularly on the (non-)existence of new particles which couple to the Higgs boson or to the top quark. Still, such processes can be exploited to probe the $\mathcal{CP}$ properties of the top-Higgs interaction, similarly to the case of single-top Higgs production [25, 26], which unlike the aforementioned processes features a Higgs boson in the final state.

On the other hand, a more direct and model-independent measurement of $y_t$ can instead be achieved by analysing the $pp \rightarrow t\bar{t}H$ production process. This approach was instrumental in providing the first evidence of the Higgs-top coupling, as reported in Refs. [7, 8]. Besides the magnitude of $y_t$, also the $\mathcal{CP}$ structure of the top-Higgs interaction has been studied at the LHC with astonishing scrutiny. Both ATLAS and CMS have reported Higgs-top $\mathcal{CP}$ studies, investigating the $pp \rightarrow ttH$ process with different Higgs boson decay channels. Although the LHC measurements support the SM $y_t$ coupling, a $\mathcal{CP}$-violating coupling has not yet been ruled out [27–32]. The presence of the latter would of course mean a departure from the SM predictions and evidence for new physics effects.

The computation of theoretical predictions including higher-order effects in perturbation theory for the $pp \rightarrow t\bar{t}H$ process with stable top quarks began two decades ago. The first computations at next-to-leading order (NLO) in QCD have been carried out by two independent groups [33–37]. About one decade later, the NLO EW corrections [38–40], and the so-called complete-NLO predictions have been computed [41]. These complete theoretical predictions comprise all leading and subleading LO contributions as well as their

corresponding higher-order QCD and EW effects. The complexity of the final state, with three massive particles, has prevented the computation of next-to-next-to-leading order (NNLO) predictions for a long time. A first step was made in Ref. [42] where NNLO QCD corrections for the flavour off-diagonal partonic channels were computed. Only recently, a complete NNLO calculation, including the diagonal partonic channels, has been presented in Ref. [43], albeit with a soft Higgs boson approximation for the two-loop amplitudes. This approach has been further improved in Ref. [44] where the soft Higgs boson approximation is combined with a high-energy expansion in the small top-mass limit. Despite substantial recent advancements, the quest for the calculation of the full two-loop amplitude for the $t\bar{t}H$ process is still ongoing, see e.g. Refs. [45–47]. In Ref. [44], the newly computed NNLO QCD results, based on the approximated double-virtual contribution, have also been equipped with the complete set of EW corrections. Before NNLO predictions became available, higher-order effects have been estimated via resummation techniques, notably for those contributions arising from soft-gluon emissions. These have been computed by different groups up to next-to-next-to-leading logarithmic (NNLL) accuracy [48–52], and have subsequently been matched to the complete-NLO predictions [53–55]. The $pp \to t\bar{t}H$ production process has also been examined in the Standard Model Effective Field Theory at the NLO level in QCD, see e.g. Refs. [56, 57].

For what concerns the simulation of various differential cross-section distributions, NLO QCD predictions for the $pp \to t\bar{t}H$ process matched to parton-shower simulations have been available for over a decade [58, 59]. More recently, the simulation of full off-shell effects at NLO QCD [60, 61] and also at NLO EW [62] have been attained. All resonant and non-resonant Feynman diagrams, interferences, and finite-width effects of the top quarks and $W^{\pm}/Z$ gauge bosons have been included in these calculations. In practice, higher-order QCD and EW corrections have been calculated for the final state $\ell^+ \nu_\ell \, \ell^- \bar{\nu}_\ell \, b\bar{b} \, H$, where $\ell^{\pm} = e^{\pm}, \mu^{\pm}$. The results presented in Ref. [61] also incorporated effects due to decays of the Higgs boson, albeit in the narrow-width approximation. In addition, NLO QCD predictions are available in the literature, based on either on-shell or full off-shell modelling of top-quark decays, which take into account the mixing between the Higgs boson's $\mathcal{CP}$-even and $\mathcal{CP}$-odd states for various observables [25, 26, 63, 64].

In this work, we provide cross-section predictions for the $pp \to t\bar{t}H$ process which represent the new state-of-the-art for what concerns the inclusion of higher-order effects. Our computation builds upon the NNLO QCD result recently presented in Ref. [44] and derived through suitable approximations of the genuine two-loop contribution. This result is supplemented by soft-gluon resummation up to NNLL accuracy and by the complete-NLO corrections. Besides the theoretical prediction, we provide a comprehensive estimate of the theoretical uncertainties due to missing higher-order effects, parton distribution functions, the strong coupling $\alpha_s$ and the top-quark mass $m_t$.

The rest of the paper is organised as follows. In section 2, we review the theoretical framework and summarise the various contributions that enter our state-of-the-art predictions. In section 3 we present our results. Our conclusions are given in section 4. Finally, in section 5 we present our citation policy, which should be followed when the results of our work are used in other scientific papers.

## 2 Theoretical predictions for $t\bar{t}H$

In this section, we discuss the technical details for the various contributions entering our predictions at NNLO+NNLL accuracy, including the complete-NLO corrections. We will also introduce the naming conventions used throughout the paper. Starting with QCD effects, in section 2.1 we report on the computation of the NNLO corrections, while in section 2.2 and section 2.3 we present the two resummation frameworks employed, namely soft-collinear effective theory (SCET) and direct QCD (dQCD). These two methods are compared in section 2.4, while the inclusion of EW effects is discussed in section 2.5. The reader not interested in the technical details may skip directly to section 2.6, where we outline the naming conventions for predictions computed at different accuracies and explain how they are combined to obtain our state-of-the-art predictions.

### 2.1 NNLO predictions

It is well known that a bottleneck in performing NNLO QCD calculations is the availability of the corresponding two-loop amplitudes. This is particularly true for processes beyond the $2 \rightarrow 2$ scattering topology, especially when several mass scales are involved. Despite the significant progress achieved in the last few years in multiloop computations (see e.g. Ref. [65] and references therein), to date, exact two-loop amplitudes for $t\bar{t}H$ production are still unavailable. For practical phenomenological applications, a promising strategy consists in obtaining the double-virtual contribution in some approximate form while keeping the rest of the NNLO calculation exact.

Besides the treatment of the two-loop amplitudes, the mediation of infrared (IR) singularities between the different amplitudes and phase spaces is also a challenging task. In the NNLO calculation of Refs. [43, 44] the transverse-momentum ($q_T$) subtraction method [66] was employed. This method uses IR subtraction counterterms that are constructed by considering the $q_T$ distribution of the produced final-state system in the limit $q_T \rightarrow 0$ [67–70]. Originally developed for the production of a colour singlet system, the method has been extended to heavy-quark production [71, 72] and applied to the NNLO computations of top-quark and bottom-quark pair production [73–75].

The production of a heavy-quark pair accompanied by a colourless particle does not pose any additional conceptual complications in the context of the $q_T$ subtraction formalism. However, its implementation requires the computation of appropriate soft-parton contributions. The results of this computation at NLO and, partly, at NNLO were presented in Ref. [42], and the evaluation of the NNLO soft terms has been subsequently completed [76] and applied to several associated heavy-quark pair production processes [43, 44, 77, 78].

Following the NNLO computation of the off-diagonal partonic channels [42], a first complete NNLO result for $t\bar{t}H$ production was presented in Ref. [43], where a purely soft Higgs boson approximation was developed to estimate the yet unknown two-loop amplitudes. A step forward was recently made in Ref. [44], where the first fully differential results for $t\bar{t}H$ production were presented and a complementary and rather different approximation of the two-loop amplitudes was introduced. This approximation is based on a high-energy (or small top-quark mass) expansion.

To isolate the part of the NNLO calculation that requires an approximation, the two-loop hard-virtual coefficient is defined as

$$H^{(2)}(\mu_{IR}) = \left. \frac{2\mathrm{Re}\left(\mathcal{M}^{(2),\mathrm{fin}}(\mu_{IR},\mu_R)\mathcal{M}^{(0)*}\right)}{|\mathcal{M}^{(0)}|^2} \right|_{\mu_R=Q}, \qquad (2.1)$$

and is computed through the interference of the Born amplitude $\mathcal{M}^{(0)}$ for the $c\bar{c} \to t\bar{t}H$ process ($c = q, g$) with the IR-subtracted two-loop amplitude $\mathcal{M}^{(2),\mathrm{fin}}(\mu_{IR},\mu_R)$ (in an expansion in powers of $\alpha_s/(2\pi)$). To be precise, $\mathcal{M}^{(2),\mathrm{fin}}(\mu_{IR},\mu_R)$ is evaluated within the scheme of Ref. [79] at the subtraction scale $\mu_{IR}$. The central value of $\mu_{IR}$ is set to the invariant mass $Q$ of the event. The ensuing contribution of the $H^{(2)}$ coefficient to the NNLO cross section reads

$$d\sigma_{H^{(2)}} \equiv \left(\frac{\alpha_s(\mu_R)}{2\pi}\right)^2 H^{(2)}(Q)\, d\sigma_{\mathrm{LO}}, \qquad (2.2)$$

where a summation over the $q\bar{q}$ and $gg$ partonic channels is left understood.

The NNLO coefficient $H^{(2)}$ is estimated by applying two independent approximations to both the numerator and denominator of Eq. (2.1). Effectively, this *reweighting* procedure corresponds to a rescaling of the approximated two-loop finite remainder by the exact Born amplitude, thus significantly improving the quality of the approximations.

The first approach relies on a *soft Higgs boson approximation* introduced for the first time in Ref. [43]. In this limit, the all-order $t\bar{t}H$ finite remainder satisfies the following factorisation formula,

$$\mathcal{M}^{\mathrm{fin}}(\{p_i\}, q; \mu_R, \mu_{IR}) \simeq F(\alpha_s(\mu_R), \mu_R/m_t)\, \frac{m_t}{v}\, \left(\frac{m_t}{p_3 \cdot q} + \frac{m_t}{p_4 \cdot q}\right) \mathcal{M}^{\mathrm{fin}}_{t\bar{t}}(\{p_i\}; \mu_R, \mu_{IR}), \qquad (2.3)$$

where $v = (\sqrt{2}G_\mu)^{-1/2}$ is the Higgs vacuum expectation value and $m_t$ the top-quark mass, $p_3$ and $p_4$ are the four-momenta of the final-state top quarks, and $q$ is the momentum of the Higgs boson. The perturbative function $F(\alpha_s(\mu_R), \mu_R/m_t)$ can be extracted by taking the soft limit of the heavy-quark scalar form factor, whose explicit expression up to $\mathcal{O}(\alpha_s^3)$ is given in Ref. [44]. With $\mathcal{M}^{\mathrm{fin}}_{t\bar{t}}$ we denote the finite remainder of the scattering amplitude for $t\bar{t}$ production, which is known up to the two-loop order [80]. In Eq. (2.3), the symbol $\simeq$ means that we have neglected contributions that are less singular than $1/q$ in the soft-Higgs limit $q \to 0$.

In the second approach, the two-loop coefficient $H^{(2)}$ is approximated in the high-energy limit ($m_t \ll Q$) of the top quarks, via the so-called *massification* procedure [81–85]. Up to power corrections in $m_t/Q$, we can write

$$\mathcal{M}^{\mathrm{fin}}(\{p_i\}, q; \mu) \simeq \mathcal{F}_{[c]}\left(\alpha_s(\mu), \frac{\mu^2}{m_t^2}, \frac{\mu^2}{2p_i \cdot p_j}\right) \mathcal{M}^{\mathrm{fin}}_{(m_t=0)}(\{p_i\}, q; \mu), \qquad (2.4)$$

where $\mu = \mu_{IR} = \mu_R$. In Eq. (2.4), $\mathcal{M}^{\mathrm{fin}}_{(m_t=0)}$ denotes the finite remainder for the production of a Higgs boson plus four massless partons, available up to the two-loop order in Ref. [86,

87], whereas $\boldsymbol{\mathcal{F}}_{[c]}$ is a perturbative process-dependent colour operator, whose expression is given in Ref. [44], and depends on the partonic channel $c = q, g$.

The two-loop contribution is ultimately obtained by combining the results from the soft (2.3) and high-energy (2.4) approximations through a weighted average. Together with the estimate of the double-virtual contribution, a procedure was worked out to quantify the systematic error due to such an approximation. More specifically, for each partonic channel, an error on $d\sigma_{H^{(2)}}$ is separately defined for the soft approximation and the massification approach. This error takes into account the relative discrepancy between the exact and approximated predictions at NLO as well as the effects on $d\sigma_{H^{(2)}}$ due to the variation of the subtraction scale $\mu_{IR}$, at which the approximations in Eqs. (2.3) and (2.4) are applied. For further details on this procedure, we refer the reader to section 3 of Ref. [44].

The two-loop contribution in Eq. (2.2), computed as discussed above, is finally combined with the remaining NNLO terms, all evaluated exactly (including the one-loop squared contribution), within the MATRIX framework [88]. In MATRIX, IR singularities are handled and cancelled via a process-independent implementation of the $q_T$-subtraction formalism extended to heavy-quark production, as mentioned above. All NLO-like singularities are treated by dipole subtraction [89–96]. The required tree-level and one-loop amplitudes are obtained via OPENLOOPS [97–99] and RECOLA [100–102].

The systematic error due to the approximation of the double-virtual contribution turns out to be sufficiently small to be subdominant with respect to the perturbative uncertainties at NNLO QCD. However, the uncertainty due to such an approximation cannot be neglected within the theory-uncertainty budget of the matched results presented in the following.

## 2.2 NNLL resummation in SCET

The approach to soft gluon emission resummation based on SCET is reviewed in this section. The resummation framework relies on the factorisation of the partonic cross section in the soft emission limit. For the case of the $pp \to t\bar{t}H$ production process, the factorisation formula is derived following closely the same procedure applied to threshold resummation in Drell-Yan [103] (for a didactic introduction, see also [104]) and top-pair production [105]. The case of the associated production of a top-pair and a Higgs boson is considered in detail in Refs. [49, 50]. The closely related cases of $t\bar{t}W$ and $t\bar{t}Z$ production are addressed in Refs. [106, 107]. The interested reader can find a more detailed discussion of the content of this section in Refs. [50, 106].

In momentum space, the parameter that regulates the soft limit is $z = Q^2/\hat{s}$, where $Q$ is the invariant mass of the $t\bar{t}H$ final state and $\sqrt{\hat{s}}$ is the partonic centre of mass energy. In the soft limit, $z \to 1$, the total cross section factorizes as follows:

$$\sigma\left(S, m_t, m_H\right) = \frac{1}{2S} \int_{\tau_{min}}^1 d\tau \int_\tau^1 \frac{dz}{\sqrt{z}} \sum_{ij} f\!f_{ij}\left(\frac{\tau}{z}, \mu\right)$$
$$\times \int d\mathrm{PS}_{t\bar{t}H} \mathrm{Tr}\left[\mathbf{H}_{ij}\left(\{p\}, \mu\right) \mathbf{S}_{ij}\left(\frac{Q(1-z)}{\sqrt{z}}, \mu\right)\right], \qquad (2.5)$$

where $\sqrt{S}$ is the hadronic centre of mass energy,

$$\tau_{\min} = \frac{(2m_t + m_H)^2}{S}, \qquad \tau = \frac{Q^2}{S}, \tag{2.6}$$

and the symbol $\{p\}$ indicates the set of the momenta of the incoming partons $i$ and $j$ together with the top quark, anti-top quark and Higgs momenta. The parton luminosity functions $f\!f_{ij}$ are defined as the convolutions of the PDFs of the $i$ and $j$ partons. In the case of $t\bar{t}H$ production, the two channels contributing to the factorisation formula in the soft limit are the quark annihilation channel and the gluon fusion channel. The phase space measure for the $t\bar{t}H$ final state is indicated by $d\mathrm{PS}_{t\bar{t}H}$. The trace $\mathrm{Tr}\,[\mathbf{HS}]$ is proportional to the spin and colour averaged partonic matrix elements for the $t\bar{t}H + X_s$ production process, where $X_s$ indicates the unobserved soft gluons in the final state. The hard functions $\mathbf{H}_{ij}$ are matrices in colour space and can be calculated starting from the colour decomposed amplitudes for the virtual corrections to the partonic tree-level $t\bar{t}H$ production diagrams. Also, the soft functions $\mathbf{S}_{ij}$ are matrices in colour space. The soft functions can be obtained from the colour decomposed real emission amplitudes evaluated in the soft limit $z \to 1$.

To perform soft-gluon resummation it is convenient to rewrite the total cross section (2.5) in terms of the Mellin transformed partonic differential cross section as

$$\sigma\left(S, m_t, m_H\right) = \frac{1}{2S} \int_{\tau_{min}}^{1} \frac{d\tau}{\tau} \frac{1}{2\pi i} \int_{c-i\infty}^{c+i\infty} \frac{dN}{\tau^N} \sum_{ij} \widetilde{f\!f}_{ij}\left(N, \mu\right) \int d\mathrm{PS}_{t\bar{t}H} d\widetilde{\sigma}_{ij}\left(N, \mu\right), \tag{2.7}$$

where the tilde indicates the Mellin transform of the luminosity function and of the partonic differential cross section. In particular, the Mellin transform of the partonic differential cross section, a.k.a. the hard scattering kernel, can be written as

$$d\widetilde{\sigma}_{ij}\left(N, \mu\right) = \mathrm{Tr}\left[\mathbf{H}_{ij}\left(\{p\}, \mu\right) \widetilde{\mathbf{s}}_{ij}\left(\ln\frac{Q^2}{\bar{N}^2 \mu^2}, \mu\right)\right]. \tag{2.8}$$

Since the soft limit $z \to 1$ corresponds to the limit $N \to \infty$ in Mellin space, terms suppressed by powers of $1/N$ were neglected in Eq. (2.7). The quantity $\bar{N}$ that appears in the Mellin transform of the soft function, $\widetilde{\mathbf{s}}$, is defined by the relation $\bar{N} = N e^{\gamma_E}$, where the Euler constant is $\gamma_E \approx 0.577216 \cdots$.

The hard $\mathbf{H}$ and soft $\widetilde{\mathbf{s}}$ functions, appearing in the hard scattering kernels in Eq. (2.8), can be evaluated in fixed-order perturbation theory at values of the scale $\mu$ at which they are free from numerically large logarithms. The scale chosen for the evaluation of the hard function is indicated by $\mu_h$ and that for the soft function is indicated by $\mu_s$. Treating $\mu_s/\mu_h \ll 1$, there is thus no common value $\mu$ which can be made to eliminate large logarithms in both functions simultaneously. This problem is circumvented by deriving renormalisation-group equations (RGEs) that can be solved to evolve the hard and soft functions from their natural scales $\mu_h$ and $\mu_s$ to a common factorisation scale $\mu_F$ at which the PDFs are evaluated. Formally, the result of this operation is [50]

$$d\widetilde{\sigma}_{ij}\left(N, \mu_F\right) = \mathrm{Tr}\left[\widetilde{\mathbf{U}}_{ij}\left(\bar{N}, \{p\}, \mu_F, \mu_h, \mu_s\right) \mathbf{H}_{ij}\left(\{p\}, \mu_h\right) \widetilde{\mathbf{U}}_{ij}^{\dagger}\left(\bar{N}, \{p\}, \mu_F, \mu_h, \mu_s\right)\right.$$
$$\left. \times \widetilde{\mathbf{s}}_{ij}\left(\ln\frac{Q^2}{\bar{N}^2 \mu_s^2}, \mu_s\right)\right]. \tag{2.9}$$

The evolution factors $\widetilde{\mathbf{U}}$, which depend on the partonic channel, resum large logarithmic corrections depending on the ratio $\mu_s/\mu_h$. They can be expressed as [106]

$$\widetilde{\mathbf{U}}_{ij}\left(\bar{N}, \{p\}, \mu_F, \mu_h, \mu_s\right) = \exp\left\{\frac{4\pi}{\alpha_s(\mu_h)}g_1\left(\lambda_s, \lambda_f\right) + g_2\left(\lambda_s, \lambda_f\right) + \frac{\alpha_s(\mu_h)}{4\pi}g_3\left(\lambda_s, \lambda_f\right) + \cdots\right\}$$
$$\times\, \mathbf{u}_{ij}\left(\{p\}, \mu_h, \mu_s\right), \tag{2.10}$$

where $\mathbf{u}$ is the non-diagonal part of the evolution matrix, and

$$\lambda_s \equiv \frac{\alpha_s(\mu_h)}{2\pi}\beta_0\ln\frac{\mu_h}{\mu_s}, \qquad \lambda_f \equiv \frac{\alpha_s(\mu_h)}{2\pi}\beta_0\ln\frac{\mu_h}{\mu_F}. \tag{2.11}$$

The functions $g_i$ depend on the cusp and PDFs anomalous dimensions. The function $g_1$ is referred to as the leading logarithmic (LL) function, the function $g_2$ is known as the next-to-leading logarithmic (NLL) function, etc.[1]

At all orders in perturbation theory, the l.h.s. of Eq. (2.9) does not depend on the choice of the hard and soft scales, $\mu_h$ and $\mu_s$. However, in practice the hard and soft functions can only be evaluated up to some finite order in perturbation theory. This fact introduces in any numerical evaluation of Eq. (2.9) a residual dependence on the choice of $\mu_h$ and $\mu_s$. The hard and soft scales are chosen such that $\mu_h \sim Q$ and $\mu_s \sim Q/\bar{N}$. While this choice of soft scale allows all large logarithms involving the Mellin parameter $N$ to be resummed, when integrating over $N$ as in Eq. (2.7) to obtain the physical cross section, one faces the well-known problem of a branch cut for large values of $N$ in the hard scattering kernel, which is related to the existence of the Landau pole in $\alpha_s$. This issue is taken care of by adopting the *Minimal Prescription* introduced in Ref. [109].

The resummed formulas include certain towers of logarithms to all orders in perturbation theory, but neglect contributions that are subleading in the soft limit. These subleading corrections can be added back in fixed-order perturbation theory through a matching procedure. For the NNLO+NNLL result in $ttH$ production, this matching procedure is implemented by evaluating the hadronic differential cross section according to

$$d\sigma_{ttH}^{\text{NNLO+NNLL}} = d\sigma_{ttH}^{\text{NNLL}} + \left(d\sigma_{ttH}^{\text{NNLO}} - d\sigma_{ttH}^{\text{NNLL}}\Big|_{\substack{\text{NNLO}\\\text{expansion}}}\right), \tag{2.12}$$

where the third term above is the NNLO expansion of the NNLL resummation formula, which is obtained by treating logarithms of scale ratios as $\mathcal{O}(1)$ quantities and re-expanding the NNLL result to the second relative order in $\alpha_s(\mu_F)$. The most non-trivial contribution is the second-order correction, which is derived in complete analogy to the top-pair production case [108] and involves a term in the Mellin-transformed partonic cross section which reads

$$d\widetilde{\hat{\sigma}}_{ij}^{(2)}\left(N, \mu_F\right) = \text{Tr}\left[\mathbf{H}_{ij}^{(2)}\left(\mu_F\right)\tilde{\mathbf{s}}_{ij}^{(0)}\left(\mu_F\right) + \mathbf{H}_{ij}^{(1)}\left(\mu_F\right)\tilde{\mathbf{s}}_{ij}^{(1)}\left(\mu_F\right) + \mathbf{H}_{ij}^{(0)}\left(\mu_F\right)\tilde{\mathbf{s}}_{ij}^{(2)}\left(\mu_F\right)\right]$$
$$- \text{Tr}\left[\mathbf{H}_{ij}^{(2)}\left(\mu_h\right)\tilde{\mathbf{s}}_{ij}^{(0)}\left(\mu_s\right) + \mathbf{H}_{ij}^{(1)}\left(\mu_h\right)\tilde{\mathbf{s}}_{ij}^{(1)}\left(\mu_s\right) + \mathbf{H}_{ij}^{(0)}\left(\mu_h\right)\tilde{\mathbf{s}}_{ij}^{(2)}\left(\mu_s\right)\right], \tag{2.13}$$

---

[1]Explicit results can be found in Appendix C.1 of [108], where the functions $g_i$ in Eq. (2.10) are denoted instead by $g_i^m$.

where the superscript $(n)$, $n = 0, 1, 2$, indicates the order in $\alpha_s$ at which the various contributions are evaluated. The formula in Eq. (2.12) for the NNLO+NNLL cross section is such that the first term takes into account the all-order soft-gluon resummation to NNLL, while the combination of the terms in parenthesis adds to it the subleading pieces in the soft limit to NNLO in fixed order.

### 2.2.1 Introduction of the renormalisation scale and study of residual scale dependence

The residual scale uncertainty affecting the phenomenological predictions presented in Refs. [50, 54] was assessed through the conventional procedure in SCET, namely by independently varying the hard, soft and factorisation scales present in the resummation formula by factors of 2 and 1/2 with respect to their default choices. Subsequently, the three types of scale variation were then added in quadrature in order to quote a total scale uncertainty, as detailed in section 3 of Ref. [50] and section 3.2 of Ref. [54].

While the approach adopted in Refs. [50, 54] is sound and reasonably conservative, it makes a direct comparison with the results in Refs. [48, 51] somewhat cumbersome. This is due to the fact that the results in Refs. [48, 51] allow to vary separately the factorisation and renormalisation scales, which are set equal in Refs. [50, 54], while in Refs. [48, 51] the soft and hard scales are kept fixed. Moreover, the uncertainties in fixed-order NNLO calculations are typically evaluated through a 7-point variation of the factorisation and renormalisation scales, so in order to compare such calculations with NNLO+NNLL results one must retain distinct factorisation and renormalisation scales also in the resummed part of the calculation.

For these reasons, the $t\bar{t}H$ cross-section predictions evaluated in this work through the method of Refs. [50, 54] are obtained after introducing the renormalisation scale $\mu_R$ in the resummation formula. This is done by eliminating $\alpha_s(\mu_h)$ in favour of $\alpha_s(\mu_R)$ by means of the relation

$$\alpha_s(\mu_h) = \frac{\alpha_s(\mu_R)}{X} \left[ 1 - \frac{\alpha_s(\mu_R)}{4\pi} \frac{\beta_1}{\beta_0} \frac{\ln X}{X} + \left( \frac{\alpha_s^2(\mu_R)}{4\pi} \right)^2 \left( \frac{\beta_1^2}{\beta_0^2} \frac{\ln^2 X - \ln X - 1 + X}{X^2} \right. \right.$$
$$\left. \left. + \frac{\beta_2}{\beta_0} \frac{1 - X}{X} \right) + \cdots \right] , \tag{2.14}$$

where

$$X = 1 - \frac{\alpha_s(\mu_R)}{2\pi} \beta_0 \ln \frac{\mu_R}{\mu_h} . \tag{2.15}$$

Once the scale $\mu_R$ has been introduced, the resummed partonic cross section is re-expanded in powers of $\alpha_s(\mu_R)$, treating logarithms of any two scale ratios as $\mathcal{O}(1)$.

It is then necessary to specify how the residual scale uncertainty affecting the predictions is obtained. In order to make comparisons with the results discussed in Refs. [53–55], the soft scale $\mu_s$ was set equal to $Q/\bar{N}$ irrespectively from the choice made for the other scales. Three different choices were made for the central values $\mu_0$ given to $\mu_F$ and $\mu_R$: $\mu_0$ was set equal to a) $\mu_0 = Q/2$, b) $\mu_0 = H_T/2$, or c) $\mu_0 = m_t + m_H/2$.

For each choice of $\mu_0$, the factorisation and renormalisation scales were varied independently by employing the usual 7-point method. In addition, a value for $\mu_h$ must be chosen. For each choice of $\mu_F$ and $\mu_R$, the cross section was evaluated both with $\mu_h = \mu_F$ and with $\mu_h = \mu_R$. In summary, for each $\mu_0$ the total cross section at NNLO+NNLL was evaluated for 11 scale choices:

$$(\mu_F/\mu_0, \mu_R/\mu_0, \mu_h/\mu_0) \in \{(1,1,1), (2,1,2), (2,1,1), (1/2,1,1/2), (1/2,1,1), (1,2,1),$$
$$(1,2,2), (1,1/2,1/2), (1,1/2,1), (2,2,2), (1/2,1/2,1/2)\}. \tag{2.16}$$

Finally, the scale uncertainty affecting the cross section is determined by taking the envelope of these 11 scale choices.[2]

## 2.3 NNLL resummation in dQCD

In this section, we describe the calculations of the NNLO+NNLL cross section for the process $pp \to t\bar{t}H$, carried out in dQCD. In this formalism, the resummation of large logarithmic corrections in the threshold limit can be achieved either by direct diagrammatic analysis [110] or all-order factorisation properties of partonic cross sections [111]. For processes with four or more coloured legs, which is the case here, the non-trivial colour flow needs to be accounted for [112, 113]. The first application of threshold resummation to calculate the $t\bar{t}H$ cross section was carried out in Ref. [48], where the process was considered at the absolute production threshold limit. Here we briefly review the calculations presented in Ref. [51], where the resummed $t\bar{t}H$ cross section was obtained at the NNLL accuracy using the threshold definition with respect to the invariant mass of the final state system. The same formalism has also been employed to obtain the NNLL predictions for the $t\bar{t}Z$ and $t\bar{t}W$ production processes [53, 55].

The resummation of logarithmic corrections which become large close to the production threshold, i.e. when the invariant mass $Q^2$ of the $t\bar{t}H$ system approaches the partonic centre of mass energy $\hat{s}$, takes place in the space of Mellin moments $N$. At the partonic level, the Mellin transform of $d\hat{\sigma}/dQ^2$ reads

$$\frac{\mathrm{d}\tilde{\hat{\sigma}}_{ij \to t\bar{t}H}}{\mathrm{d}Q^2}(N, Q^2, \{m^2\}, \mu_R^2, \mu_F^2) = \int_0^1 \mathrm{d}\hat{\rho}\, \hat{\rho}^{N-1} \frac{\mathrm{d}\hat{\sigma}_{ij \to t\bar{t}H}}{\mathrm{d}Q^2}(\hat{\rho}, Q^2, \{m^2\}, \mu_R^2, \mu_F^2), \tag{2.17}$$

where $\hat{\rho} = Q^2/\hat{s}$, $\{m^2\}$ stands for all masses entering the calculations and $i, j$ denote two initial-state coloured partons. The cross section factorises into a product of Mellin-transformed functions of the coupling constant and the ratios of the scales,

$$\frac{d\tilde{\hat{\sigma}}_{ij \to t\bar{t}H}^{(\text{NNLL})}}{dQ^2}(N, Q^2, \{m^2\}, \mu_R^2, \mu_F^2) = \mathrm{Tr}\left[\mathbf{H}_R(Q^2, \{m^2\}, \mu_R^2, \mu_F^2)\right.$$
$$\times \bar{\mathbf{U}}_R(N+1, Q^2, \{m^2\}, \mu_R^2)\tilde{\mathbf{S}}_R(N+1, Q^2, \{m^2\})\,\mathbf{U}_R(N+1, Q^2, \{m^2\}, \mu_R^2)\Big]$$
$$\times \Delta^i(N+1, Q^2, \mu_R^2, \mu_F^2)\,\Delta^j(N+1, Q^2, \mu_R^2, \mu_F^2), \tag{2.18}$$

---

[2]While $\mu_s = Q/\bar{N}$ is kept fixed across these 11 scale choices, we verified that varying it up and down by a factor of 2 for $\mu_h = \mu_F = \mu_R = \mu_0$ does not change the uncertainty envelopes for results obtained in this paper.

where $\mathbf{H}_R$, $\bar{\mathbf{U}}_R$, $\mathbf{U}_R$ and $\tilde{\mathbf{S}}_R$ are matrices in colour space over which the trace is taken. The jet functions $\Delta^i$ account for (soft-)collinear logarithmic contributions from the initial state partons and are well known at NNLL [109, 114]. The term $\bar{\mathbf{U}}_R \tilde{\mathbf{S}}_R \mathbf{U}_R$ originates from a solution of the renormalisation group equation for the soft function and consists of the evolution matrices $\bar{\mathbf{U}}_R$, $\mathbf{U}_R$, as well as the function $\tilde{\mathbf{S}}_R$ which plays the role of a boundary condition of the renormalisation group equation. The evolution matrices are given by (reverse in the case of $\bar{\mathbf{U}}_R$) path-ordered exponentials of the soft anomalous dimension matrix $\bar{\mathbf{\Gamma}}_{ij\to t\bar{t}H}(\alpha_s) = \left(\frac{\alpha_s}{\pi}\right) \bar{\mathbf{\Gamma}}^{(1)}_{ij\to t\bar{t}H} + \left(\frac{\alpha_s}{\pi}\right)^2 \bar{\mathbf{\Gamma}}^{(2)}_{ij\to t\bar{t}H} + \dots$ which is obtained by subtracting the contributions already taken into account in $\Delta^i \Delta^j$ from the full soft anomalous dimension for the process $ij \to t\bar{t}H$. At NLL, the path-ordered exponentials collapse to standard exponential factors in the colour space $\mathbf{R}$ where $\mathbf{\Gamma}^{(1)}_R$ is diagonal. At NNLL, the path-ordered exponentials are eliminated by treating $\mathbf{U}_R$ and $\bar{\mathbf{U}}_R$ perturbatively

$$\mathbf{U}_R(N, Q^2, \{m^2\}, \mu_R^2) = \left(1 + \frac{\alpha_s(\mu_R^2)}{\pi[1 - 2\alpha_s(\mu_R^2)b_0 \log N]}\mathbf{K}\right)\left[e^{g_s(N)\overrightarrow{\gamma}^{(1)}}\right]_D \left(1 - \frac{\alpha_s(\mu_R^2)}{\pi}\mathbf{K}\right),$$
$$(2.19)$$

where $\overrightarrow{\gamma}^{(1)}$ is a vector of $\mathbf{\Gamma}^{(1)}_{ij\to klB}$ eigenvalues and subscript $D$ indicates a diagonal matrix. Furthermore, $\mathbf{K}_{IJ} = \delta_{IJ}\gamma_I^{(1)} \frac{b_1}{2b_0^2} - \frac{\left(\mathbf{\Gamma}^{(2)}_R\right)_{IJ}}{2\pi b_0 + \gamma_I^{(1)} - \gamma_J^{(1)}}$ with $b_0$ and $b_1$ denoting the first two $\beta_{\mathrm{QCD}}$ coefficients and

$$g_s(N) = \frac{1}{2\pi b_0}\left\{\log(1 - 2\lambda) + \alpha_s(\mu_R^2)\left[\frac{b_1}{b_0}\frac{\log(1 - 2\lambda)}{1 - 2\lambda} - 2\gamma_{\mathrm{E}}b_0\frac{2\lambda}{1 - 2\lambda}\right.\right.$$
$$\left.\left. + b_0 \log\left(\frac{Q^2}{\mu_R^2}\right)\frac{2\lambda}{1 - 2\lambda}\right]\right\} \qquad (2.20)$$

with $\lambda = \alpha_s(\mu_R^2)b_0 \log N$. The remaining function in Eq. (2.18), $\mathbf{H}_R$, contains information on the hard off-shell dynamics and collects contributions non-logarithmic in $N$ which are projected on the $\mathbf{R}$ colour basis. At NNLL, the $\mathcal{O}(\alpha_s)$ terms in the perturbative expansion of $\mathbf{H}_R$ and $\tilde{\mathbf{S}}_R$, as well as $\mathbf{\Gamma}^{(2)}_R$ are needed. While the latter is known analytically [115], the contributions of the other two functions need to be determined. In particular, the virtual corrections which enter $\mathbf{H}^{(1)}_R$ are extracted numerically from the NLO calculations by `MadGraph5_aMC@NLO` [116].

The threshold-resummed NNLL cross sections are then matched to the NNLO predictions of Ref. [44] according to

$$\frac{d\sigma^{(\mathrm{NNLO+NNLL})}_{pp\to t\bar{t}H}}{dQ^2} = \frac{d\sigma^{(\mathrm{NNLO})}_{pp\to t\bar{t}H}}{dQ^2} + \frac{d\sigma^{(\mathrm{res-exp})}_{pp\to t\bar{t}H}}{dQ^2} \qquad (2.21)$$

with

$$\frac{d\sigma^{(\mathrm{res-exp})}_{pp\to t\bar{t}H}}{dQ^2}(Q^2, \{m^2\}, \mu_R^2, \mu_F^2) = \sum_{i,j}\int_{\mathsf{C}}\frac{dN}{2\pi i}\,\rho^{-N}f^{(N+1)}_{i/h_1}(\mu_F^2)\,f^{(N+1)}_{j/h_2}(\mu_F^2)$$
$$\times\left[\frac{d\tilde{\tilde{\sigma}}^{(\mathrm{NNLL})}_{ij\to t\bar{t}H}}{dQ^2}(N, Q^2, \{m^2\}, \mu_R^2, \mu_F^2) - \frac{d\tilde{\tilde{\sigma}}^{(\mathrm{NNLL})}_{ij\to t\bar{t}H}}{dQ^2}(N, Q^2, \{m^2\}, \mu_R^2, \mu_F^2)|_{(\mathrm{NNLO})}\right], \quad (2.22)$$

where $f_{i/h}(x, \mu_F^2)$ are moments of the parton distribution functions and $d\hat{\sigma}_{ij \to t\bar{t}H}^{(\text{res})}/dQ^2 \big|_{(\text{NNLO})}$ represents the perturbative expansion of the resummed cross section truncated at NNLO. The inverse Mellin transform (2.22) is evaluated numerically according to the "Minimal Prescription" [109] along a contour $\mathsf{C}$ in the complex-$N$ space. For more information on the theoretical framework, we refer the reader to Refs. [51, 53, 55].

## 2.4   Comparison of the two resummation approaches

Although the formulas for the resummed $t\bar{t}H$ cross sections presented in the previous two subsections are derived in conceptually distinct frameworks and look quite different at first glance, the two resummation formalisms, describing the same physics in the soft gluon emission limit, are theoretically equivalent. In particular, both make use of factorisation in the soft limit along with RG-improved perturbation theory to resum logarithmic corrections, and if the formulas were evaluated to infinite logarithmic accuracy, they would agree exactly. It needs to be stressed, however, that the derivation of the two formulas is performed in very different ways. In the case of dQCD, the formalism is derived directly from the properties of scattering amplitudes in full QCD, while in SCET, effective field theory techniques are used in intermediate steps. The two distinct theoretical frameworks lead naturally to different organisations of the resummed expressions, so that when evaluated at a fixed logarithmic accuracy, the analytic and numerical results are no longer the same. Before exploring numerical results, we first highlight some of the salient differences in the analytic expressions.

In the context of the present work, a particularly noticeable difference between the two formalisms is the set of scales that are allowed to vary and the parametric counting of logarithms underlying RG-improved perturbation theory and thus resummation. In dQCD, one can vary the two scales $\mu_F$ and $\mu_R$, and $\lambda = \alpha_s(\mu_R^2)\, b_0 \ln N$ is considered an order one parameter. In SCET, on the other hand, as explained in section 2.2, the set of scales $\mu_i \in \{\mu_F, \mu_R, \mu_h\}$ is allowed to vary, and the ratio of any of these scales with each other or with $\mu_s = Q/\bar{N}$ is considered a large logarithm. As a result, while in dQCD expansion coefficients such as Eq. (2.20) depend only on $\lambda$, expansion coefficients in SCET can depend on several order-one parameters – for example, the expansion coefficients $g_i$ in Eq. (2.10) depend on $\lambda_s$ and $\lambda_f$ in Eq. (2.11), for the case $\mu_h = \mu_R$. Other notable differences include the fact that the exponential factors in dQCD (but not in SCET) are chosen to vanish in the limit $\lambda \to 0$ (see e.g. Eq. (2.20)), which is achieved by re-expanding the $\lambda$-independent terms and absorbing them into the hard function,[3] while in SCET (but not dQCD) the approximation $\exp(\alpha_s g_3) \approx 1 + \alpha_s g_3$ is used.

We have checked analytically that when exactly the same implementation of RG-improved perturbation theory is used, the SCET and dQCD formulas agree at NNLL. For numerical evaluations we have retained the differences in the two setups as outlined above, so that the SCET formulas contain some corrections that are considered $\text{N}^3\text{LL}$ and higher order (and thus not necessarily included) in the dQCD formalism and vice versa. The

---

[3]Note that due to this and other similar manipulations, the dQCD hard functions $\mathbf{H}_R$ are not identical to the SCET hard functions $\mathbf{H}$. For the same reason, contributions involving the factor $\gamma_E$ differ by terms considered $\text{N}^3\text{LL}$ and higher orders in dQCD.

numerical differences between the predictions obtained within the two formalisms can then be used as an additional handle on theoretical uncertainties in the soft gluon resummation formulas, beyond those estimated through scale variations in either approach alone. In particular, these differences can be seen as an indicator of the size of subleading terms beyond the formal accuracy of resummation, i.e. $N^3LL$ and higher.

The comparison of NNLO+NNLL results for the total cross section in SCET and dQCD is shown on the left-hand side of figure 1. The results are shown for three different parametric choices for the default values of $\mu_F$ and $\mu_R$:

- $\mu_F = \mu_R = m_t + m_H/2$

- $\mu_F = \mu_R = H_T/2$

- $\mu_F = \mu_R \equiv Q/2$ ($Q \equiv M_{ttH}$)

Scale uncertainties in SCET are obtained by evaluating the cross section for the 11 different values for $\mu_F, \mu_R$ and $\mu_h$ listed in Eq. (2.16), while those in dQCD are evaluated using the standard 7-point method. We observe that the NNLO+NNLL results agree remarkably well, with the central values differing only by a few permille.

In order to take the small differences between the two approaches as an additional theoretical uncertainty, we combine dQCD and SCET results by averaging the central values and taking the envelope of the uncertainty bands. The result of this combination is shown on the right-hand side of figure 1, where we also display the NNLO QCD results with uncertainties obtained via the 7-point method. Comparing the two sets of results, one sees that combined NNLO+NNLL predictions have not only smaller scale variation errors but are also more stable with respect to the choice of the default values of $\mu_F = \mu_R$ than the NNLO results. One also sees that the resummation effects are smallest for the default choice $\mu_F = \mu_R = m_t + m_H/2$, which gives an additional motivation for using this choice when compiling final results in section 3 (apart from the fact that this is the only physical scale available for total cross section, for which the values of dynamic scales have been integrated over).

## 2.5 NLO EW and photon-induced contributions

In this section, we briefly outline the structure of the contributions that enter the EW corrections to $t\bar{t}H$. The notation is the same as used in Refs. [38, 40, 41, 54, 117].

A given observable $\Sigma^{t\bar{t}H}$ for the process $pp \to t\bar{t}H(+X)$ can be simultaneously expanded in the QCD and EW couplings as:

$$\Sigma^{t\bar{t}H}(\alpha_s, \alpha) = \sum_{m+n \geq 3} \alpha_s^m \alpha^n \Sigma_{m+n,n} \,. \tag{2.23}$$

The LO ($m + n = 3$), NLO ($m + n = 4$) and NNLO ($m + n = 5$) contributions correspond

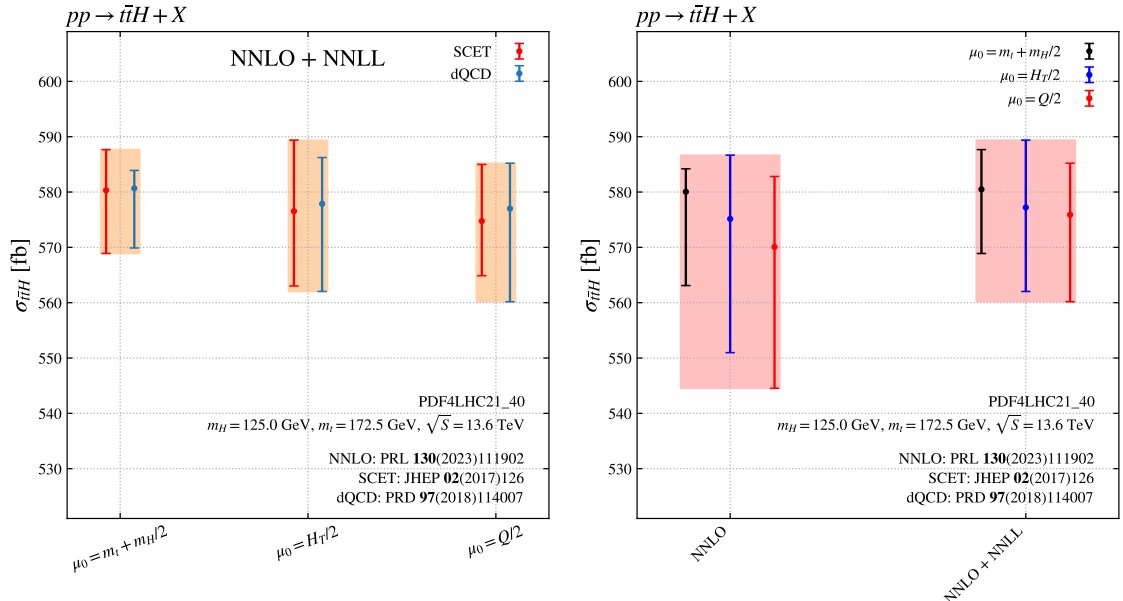

**Figure 1**: Left: comparison between NNLO+NNLL results in dQCD and SCET for three parametrically different choices of the default scales. Right: comparison of the combined NNLO+NNLL results with NNLO for the same three sets of scales. No EW corrections are included. See the text for additional explanations on the estimation of the uncertainties.

therefore to

$$\begin{aligned}
\Sigma_{\text{LO}}^{t\bar{t}H}(\alpha_s, \alpha) &= \alpha_s^2 \alpha \Sigma_{3,1} + \alpha_s \alpha^2 \Sigma_{3,2} + \alpha^3 \Sigma_{3,3} \\
&\equiv \Sigma_{\text{LO},1} + \Sigma_{\text{LO},2} + \Sigma_{\text{LO},3}, \\
\Sigma_{\text{NLO}}^{t\bar{t}H}(\alpha_s, \alpha) &= \alpha_s^3 \alpha \Sigma_{4,1} + \alpha_s^2 \alpha^2 \Sigma_{4,2} + \alpha_s \alpha^3 \Sigma_{4,3} + \alpha^4 \Sigma_{4,4} \\
&\equiv \Sigma_{\text{NLO},1} + \Sigma_{\text{NLO},2} + \Sigma_{\text{NLO},3} + \Sigma_{\text{NLO},4}, \\
\Sigma_{\text{NNLO}}^{t\bar{t}H}(\alpha_s, \alpha) &= \alpha_s^4 \alpha \Sigma_{5,1} + \alpha_s^3 \alpha^2 \Sigma_{5,2} + \alpha_s^2 \alpha^3 \Sigma_{5,3} + \alpha_s \alpha^4 \Sigma_{5,4} + \alpha^5 \Sigma_{5,5} \\
&\equiv \Sigma_{\text{NNLO},1} + \Sigma_{\text{NNLO},2} + \Sigma_{\text{NNLO},3} + \Sigma_{\text{NNLO},4} + \Sigma_{\text{NNLO},5}.
\end{aligned} \tag{2.24}$$

The contributions $\Sigma_{\text{LO},1}$, $\Sigma_{\text{NLO},1}$, and $\Sigma_{\text{NNLO},1}$ are usually referred to as the LO contribution to the $t\bar{t}H$ cross section, and its NLO and NNLO corrections in QCD; the quantity $\Sigma_{\text{NLO},2}$ is usually referred to as the NLO EW corrections. Finally, a prediction including all LO and NLO contributions is said to be computed at complete-NLO accuracy. We will neglect NNLO contributions different than $\Sigma_{\text{NNLO},1}$.

LO and NLO contributions different from $\Sigma_{\text{LO},1}$ and $\Sigma_{\text{NLO},1}$ can involve partonic processes with at least one photon in the initial state and therefore depend on the photon PDF. The dominant contribution originates from the process $g\gamma \to t\bar{t}H$,[4] which enters both LO and NLO. However, also $q\gamma$ and $\gamma\gamma$ initial states are possible. The quantities $\Sigma_{\text{NLO EW}}$, $\Sigma_{\text{NLO},3}$ and $\Sigma_{\text{NLO},4}$ receive contributions from the $q\gamma \to t\bar{t}Hq$ processes, while

---

[4]See Ref. [118] for an analogous and more detailed discussion for the case of $t\bar{t}$ production.

the $\gamma\gamma$ initial state contributes to $\Sigma_{\text{LO},3}$, via $\gamma\gamma \to t\bar{t}H$, to $\Sigma_{\text{NLO},3}$, via $\gamma\gamma \to t\bar{t}Hg$, and to $\Sigma_{\text{NLO},4}$, via $\gamma\gamma \to t\bar{t}H\gamma$.

## 2.6 Naming convention and construction of the state-of-the-art predictions

Having discussed all the theoretical ingredients entering the $t\bar{t}H$ cross section, we now introduce the naming convention for the quantities presented in the remainder of this work. We focus on the total cross section $\sigma$ and, owing to the dominance of QCD-type effects, we call

$$\sigma_{\text{LO}} \equiv \sigma_{\text{LO},1} , \tag{2.25}$$

$$\sigma_{\text{NLO}} \equiv \sigma_{\text{LO},1} + \sigma_{\text{NLO},1} , \tag{2.26}$$

$$\sigma_{\text{NNLO}} \equiv \sigma_{\text{LO},1} + \sigma_{\text{NLO},1} + \sigma_{\text{NNLO},1} . \tag{2.27}$$

These fixed-order predictions, in particular $\sigma_{\text{NNLO}}$, can be supplemented with soft-gluon resummation (we consider only NNLL accuracy for simplicity). If we call $\sigma_{\text{NNLL}}^{\text{SCET}}$ ($\sigma_{\text{NNLL}}^{\text{dQCD}}$) the purely resummed prediction at NNLL accuracy obtained with SCET (dQCD), the corresponding matched predictions are then obtained by adding these quantities to $\sigma_{\text{NNLO}}$, and removing the double counting,

$$\sigma_{\text{NNLO+NNLL}}^{\text{SCET}} = \sigma_{\text{NNLO}} + \sigma_{\text{NNLL}}^{\text{SCET}} - \sigma_{\text{NNLL}}^{\text{SCET}}\big|_{\alpha_s^2} , \tag{2.28}$$

$$\sigma_{\text{NNLO+NNLL}}^{\text{dQCD}} = \sigma_{\text{NNLO}} + \sigma_{\text{NNLL}}^{\text{dQCD}} - \sigma_{\text{NNLL}}^{\text{dQCD}}\big|_{\alpha_s^2} , \tag{2.29}$$

where $\sigma_{\text{NNLL}}^{\text{SCET}}\big|_{\alpha_s^2}$ ($\sigma_{\text{NNLL}}^{\text{dQCD}}\big|_{\alpha_s^2}$) is the expansion of $\sigma_{\text{NNLL}}^{\text{SCET}}$ ($\sigma_{\text{NNLL}}^{\text{dQCD}}$) up to relative order $\alpha_s^2$. The two matched predictions are combined by simply taking their arithmetic average,

$$\sigma_{\text{NNLO+NNLL}} = \frac{\sigma_{\text{NNLO+NNLL}}^{\text{SCET}} + \sigma_{\text{NNLO+NNLL}}^{\text{dQCD}}}{2} . \tag{2.30}$$

Finally, the addition of $_{\text{+EW}}$ in the subscript corresponds to including the subleading LO and NLO contributions, i.e. to the combination of the matched prediction with the complete-NLO corrections. So, for example:

$$\sigma_{\text{NNLO+EW}} = \sigma_{\text{NNLO}} + \sum_{i=2}^{3} \sigma_{\text{LO},i} + \sum_{j=2}^{4} \sigma_{\text{NLO},j} , \tag{2.31}$$

$$\sigma_{\text{NNLO+NNLL+EW}} = \sigma_{\text{NNLO+NNLL}} + \sum_{i=2}^{3} \sigma_{\text{LO},i} + \sum_{j=2}^{4} \sigma_{\text{NLO},j} , \tag{2.32}$$

and so on.

## 3 Numerical results

In this section, we provide the state-of-the-art predictions for $t\bar{t}H$. In particular, the relevant input parameters are listed in section 3.1, while the impact of the various contributions to the $t\bar{t}H$ cross section is discussed in section 3.2. In section 3.3 we estimate different sources of theoretical errors.

## 3.1  Input parameters

The input parameters for the theoretical predictions follow the recommendations of the LHC Higgs Working Group. [5] In particular, we work in the five-flavour scheme, where the top quark is the only massive fermion. The top-quark, $Z$-, and $W$-boson masses are set to

$$m_t = 172.5 \text{ GeV}, \qquad m_W = 80.379 \text{ GeV}, \qquad m_Z = 91.1876 \text{ GeV}. \qquad (3.1)$$

Vector boson masses as well as the top quark mass and Yukawa coupling are renormalised in the on-shell scheme. All particles are considered stable, and their widths are therefore neglected.

The value of the Fermi constant,

$$G_\mu = 1.16637 \times 10^{-5} \text{ GeV}^{-2}, \qquad (3.2)$$

fixes the EW input scheme. The Higgs boson mass is varied in the set of values

$$m_H \in \{124.6, 125, 125.09, 125.38, 125.6, 126\} \text{ GeV}, \qquad (3.3)$$

while three scenarios are considered for the centre-of-mass energy $\sqrt{S}$:

$$\sqrt{S} \in \{13, 13.6, 14\} \text{ TeV}. \qquad (3.4)$$

The PDF4LHC21 parton-distribution functions (PDFs) are employed [119] for all coloured partons. Specifically, we employed the `PDF4LHC21_40_pdfas` set which makes it possible to estimate the PDF-related uncertainties (using the Hessian method) together with those associated with $\alpha_s$. Regarding the photon density, which is relevant for the EW corrections to $t\bar{t}H$, a specific choice needs to be made, as it is not included in the PDF4LHC21 combination. In particular, the photon density prediction, based on the LuxQED method [120, 121], applied on top of the PDF4LHC15 combination [122] is employed. [6]

The central value $\mu$ of the renormalisation and factorisation scales is fixed to half the threshold energy:

$$\mu = \frac{m_H}{2} + m_t. \qquad (3.5)$$

The scale-uncertainty error is obtained by varying the two scales by a factor of 2, keeping $0.5 \leq \mu_R/\mu_F \leq 2$ (7-point variations).

## 3.2  Cross-section predictions, and impact of the various contributions

Before presenting the final results, it is worth considering the different contributions that enter the cross section. Starting from those contributions that are included in fixed-order perturbation theory, we identify in table 1 the impact of NNLO predictions and of EW corrections. Specifically, we define

$$\delta_{\text{NNLO}} = \frac{\sigma_{\text{NNLO}}}{\sigma_{\text{NLO}}} - 1, \qquad (3.6)$$

$$\delta_{\text{NNLO+EW}} = \frac{\sigma_{\text{NNLO+EW}}}{\sigma_{\text{NLO}}} - 1, \qquad (3.7)$$

---

[5]See https://twiki.cern.ch/twiki/bin/view/LHCPhysics/LHCHWG136TeVxsec.

[6]The possibility to apply two different sets of PDFs is achieved through PineAppl [123] and its interface to Matrix [88].

| $\sqrt{S}$ [TeV] | $m_H$ [GeV] | $\sigma_{\mathrm{NLO}}$ [fb]$^{+[\%]}_{-[\%]}$ | $\sigma_{\mathrm{NNLO}}$ [fb]$^{+[\%]}_{-[\%]}$ | $\sigma_{\mathrm{NNLO+EW}}$ [fb]$^{+[\%]}_{-[\%]}$ | $\delta_{\mathrm{NNLO}}$ [%] | $\delta_{\mathrm{NNLO+EW}}$ [%] |
|---|---|---|---|---|---|---|
| 13.0 | 124.60 | $501.7^{+5.8}_{-9.1}$ | $522.8^{+0.9}_{-3.1}$ | $533.8^{+1.1}_{-3.2}$ | 4.2 | 6.4 |
| 13.0 | 125.00 | $497.2^{+5.8}_{-9.1}$ | $519.4^{+1.0}_{-3.2}$ | $530.2^{+1.2}_{-3.3}$ | 4.5 | 6.6 |
| 13.0 | 125.09 | $496.2^{+5.8}_{-9.1}$ | $517.6^{+0.9}_{-3.1}$ | $528.4^{+1.1}_{-3.2}$ | 4.3 | 6.5 |
| 13.0 | 125.38 | $493.0^{+5.8}_{-9.1}$ | $513.8^{+0.9}_{-3.1}$ | $524.5^{+1.1}_{-3.2}$ | 4.2 | 6.4 |
| 13.0 | 125.60 | $490.5^{+5.8}_{-9.1}$ | $511.0^{+0.9}_{-3.1}$ | $521.7^{+1.1}_{-3.2}$ | 4.2 | 6.4 |
| 13.0 | 126.00 | $486.1^{+5.8}_{-9.1}$ | $506.7^{+0.9}_{-3.1}$ | $517.2^{+1.1}_{-3.2}$ | 4.2 | 6.4 |
| 13.6 | 124.60 | $563.7^{+5.9}_{-9.1}$ | $586.7^{+0.8}_{-3.0}$ | $598.9^{+1.0}_{-3.1}$ | 4.1 | 6.2 |
| 13.6 | 125.00 | $558.6^{+5.9}_{-9.1}$ | $580.1^{+0.7}_{-2.9}$ | $592.0^{+0.9}_{-3.1}$ | 3.8 | 6.0 |
| 13.6 | 125.09 | $557.5^{+5.9}_{-9.1}$ | $579.7^{+0.8}_{-3.0}$ | $591.7^{+1.0}_{-3.1}$ | 4.0 | 6.1 |
| 13.6 | 125.38 | $553.9^{+5.9}_{-9.1}$ | $576.5^{+0.8}_{-3.0}$ | $588.4^{+1.0}_{-3.1}$ | 4.1 | 6.2 |
| 13.6 | 125.60 | $551.1^{+5.9}_{-9.1}$ | $573.9^{+0.9}_{-3.0}$ | $585.6^{+1.0}_{-3.2}$ | 4.1 | 6.3 |
| 13.6 | 126.00 | $546.2^{+5.9}_{-9.1}$ | $568.5^{+0.9}_{-3.0}$ | $580.1^{+1.1}_{-3.2}$ | 4.1 | 6.2 |
| 14.0 | 124.60 | $607.0^{+6.0}_{-9.1}$ | $629.1^{+0.6}_{-2.9}$ | $642.1^{+0.8}_{-3.0}$ | 3.6 | 5.8 |
| 14.0 | 125.00 | $601.6^{+6.0}_{-9.1}$ | $625.6^{+0.8}_{-3.0}$ | $638.4^{+0.9}_{-3.1}$ | 4.0 | 6.1 |
| 14.0 | 125.09 | $600.4^{+6.0}_{-9.1}$ | $622.9^{+0.7}_{-2.9}$ | $635.6^{+0.9}_{-3.0}$ | 3.7 | 5.9 |
| 14.0 | 125.38 | $596.5^{+6.0}_{-9.1}$ | $621.1^{+0.8}_{-3.0}$ | $634.6^{+1.0}_{-3.2}$ | 4.1 | 6.4 |
| 14.0 | 125.60 | $593.6^{+6.0}_{-9.1}$ | $617.7^{+0.8}_{-3.0}$ | $630.2^{+1.0}_{-3.1}$ | 4.1 | 6.2 |
| 14.0 | 126.00 | $588.3^{+6.0}_{-9.1}$ | $611.2^{+0.7}_{-3.0}$ | $623.6^{+0.9}_{-3.1}$ | 3.9 | 6.0 |

**Table 1**: Predictions for the process $t\bar{t}H$: contributions entering the fixed-order cross section. The quoted uncertainties are obtained via 7-point scale variations.

i.e. the impact, relative to the NLO QCD predictions, of the NNLO corrections alone or combined with the complete-NLO corrections. From the table, we observe that the impact of NNLO and EW corrections is roughly independent of the Higgs mass and collider energy, and that both quantities amount to a few percent ($\sim 4\%$ for the NNLO QCD, 2% for the EW). Furthermore, the inclusion of NNLO corrections significantly reduces the theoretical uncertainties estimated from scale variations compared to NLO, shrinking them by roughly a factor of 3, down to $\sim 3\%$ when the largest variation is considered. As expected, given their small size, EW corrections have a marginal effect on the scale-variation band.

The effects of resummation can be seen by examining table 2. We note that, regardless of the framework, resummation changes the central prediction by only one per mille or below, compared to NNLO, for the choice of the scale in Eq. (3.5). However, its effect on the scale dependence is substantial: the inclusion of NNLL resummation in the prediction further reduces the scale uncertainty from 3% at NNLO down to the level of 1.5-2%.[7]

---

[7]Note that the improved stability of the NNLO+NNLL results under scale variations is more apparent when considered across a wider range of (parametrically) different scales, as in figure 1.

| $\sqrt{S}$ [TeV] | $m_H$ [GeV] | $\sigma_{\mathrm{NNLO+EW}}$ [fb]$^{+[\%]}_{-[\%]}$ | $\sigma^{\mathrm{dQCD}}_{\mathrm{NNLO+NNLL+EW}}$ [fb]$^{+[\%]}_{-[\%]}$ | $\sigma^{\mathrm{SCET}}_{\mathrm{NNLO+NNLL+EW}}$ [fb]$^{+[\%]}_{-[\%]}$ | $\sigma_{\mathrm{NNLO+NNLL+EW}}$ [fb]$^{+[\%]}_{-[\%]}$ |
|---|---|---|---|---|---|
| 13.0 | 124.60 | $533.8^{+1.1}_{-3.2}$ | $534.4^{+0.6}_{-2.1}$ | $534.1^{+0.5\ +1.6}_{-2.2\ -1.7}$ | $534.2^{+1.6}_{-2.2}$ |
| 13.0 | 125.00 | $530.2^{+1.2}_{-3.3}$ | $530.8^{+0.6}_{-2.1}$ | $530.5^{+0.7\ +1.7}_{-2.2\ -1.7}$ | $530.6^{+1.7}_{-2.3}$ |
| 13.0 | 125.09 | $528.4^{+1.1}_{-3.2}$ | $529.0^{+0.6}_{-2.1}$ | $528.7^{+0.5\ +1.6}_{-2.2\ -1.7}$ | $528.8^{+1.6}_{-2.2}$ |
| 13.0 | 125.38 | $524.5^{+1.1}_{-3.2}$ | $525.1^{+0.6}_{-2.1}$ | $524.8^{+0.6\ +1.6}_{-2.2\ -1.7}$ | $524.9^{+1.6}_{-2.2}$ |
| 13.0 | 125.60 | $521.7^{+1.1}_{-3.2}$ | $522.2^{+0.6}_{-2.1}$ | $521.9^{+0.5\ +1.6}_{-2.2\ -1.6}$ | $522.1^{+1.6}_{-2.2}$ |
| 13.0 | 126.00 | $517.2^{+1.1}_{-3.2}$ | $517.7^{+0.6}_{-2.1}$ | $517.5^{+0.5\ +1.6}_{-2.2\ -1.7}$ | $517.6^{+1.6}_{-2.2}$ |
| 13.6 | 124.60 | $598.9^{+1.0}_{-3.1}$ | $599.5^{+0.5}_{-2.0}$ | $599.2^{+0.3\ +1.5}_{-2.2\ -1.6}$ | $599.3^{+1.5}_{-2.2}$ |
| 13.6 | 125.00 | $592.0^{+0.9}_{-3.1}$ | $592.7^{+0.5}_{-2.0}$ | $592.3^{+0.2\ +1.5}_{-2.1\ -1.5}$ | $592.5^{+1.4}_{-2.1}$ |
| 13.6 | 125.09 | $591.7^{+1.0}_{-3.1}$ | $592.3^{+0.5}_{-2.0}$ | $592.0^{+0.2\ +1.5}_{-2.1\ -1.6}$ | $592.1^{+1.5}_{-2.2}$ |
| 13.6 | 125.38 | $588.4^{+1.0}_{-3.1}$ | $589.0^{+0.5}_{-2.0}$ | $588.6^{+0.3\ +1.6}_{-2.2\ -1.6}$ | $588.8^{+1.5}_{-2.2}$ |
| 13.6 | 125.60 | $585.6^{+1.0}_{-3.2}$ | $586.2^{+0.5}_{-2.1}$ | $585.9^{+0.4\ +1.6}_{-2.2\ -1.6}$ | $586.0^{+1.6}_{-2.2}$ |
| 13.6 | 126.00 | $580.1^{+1.1}_{-3.2}$ | $580.7^{+0.6}_{-2.1}$ | $580.4^{+0.4\ +1.6}_{-2.2\ -1.7}$ | $580.5^{+1.6}_{-2.2}$ |
| 14.0 | 124.60 | $642.1^{+0.8}_{-3.0}$ | $642.7^{+0.5}_{-1.9}$ | $642.4^{+0.2\ +1.3}_{-2.1\ -1.5}$ | $642.6^{+1.3}_{-2.1}$ |
| 14.0 | 125.00 | $638.4^{+0.9}_{-3.1}$ | $639.0^{+0.5}_{-2.0}$ | $638.6^{+0.3\ +1.5}_{-2.1\ -1.6}$ | $638.8^{+1.5}_{-2.2}$ |
| 14.0 | 125.09 | $635.6^{+0.9}_{-3.0}$ | $636.3^{+0.5}_{-2.0}$ | $635.9^{+0.2\ +1.4}_{-2.1\ -1.6}$ | $636.1^{+1.4}_{-2.1}$ |
| 14.0 | 125.38 | $634.6^{+1.0}_{-3.2}$ | $635.3^{+0.5}_{-2.1}$ | $634.9^{+0.3\ +1.6}_{-2.2\ -1.7}$ | $635.1^{+1.6}_{-2.2}$ |
| 14.0 | 125.60 | $630.2^{+1.0}_{-3.1}$ | $630.9^{+0.5}_{-2.1}$ | $630.4^{+0.3\ +1.6}_{-2.2\ -1.6}$ | $630.7^{+1.5}_{-2.2}$ |
| 14.0 | 126.00 | $623.6^{+0.9}_{-3.1}$ | $624.2^{+0.5}_{-2.0}$ | $623.9^{+0.3\ +1.5}_{-2.2\ -1.6}$ | $624.1^{+1.5}_{-2.2}$ |

**Table 2**: Predictions for the process $t\bar{t}H$. The quoted uncertainties are obtained via 7-point scale variations. For the SCET predictions, two such bands are quoted, respectively with $\mu_h = \mu_R$ and $\mu_h = \mu_F$.

We also see that the dQCD and the SCET prediction[8] computed with $\mu_h = \mu_R$ display a very similar, but asymmetric, scale dependence, whereas for the SCET prediction with $\mu_h = \mu_F$ scale variations are more symmetric, although the overall uncertainty band stays roughly the same. In order to combine the two resummed calculations into a single result, we average the central values as in Eq. (2.30) and take as the uncertainty band the envelope of scale variations across the two methods. The combined prediction obtained in this way, which we denote as $\sigma_{\mathrm{NNLO+NNLL+EW}}$, is shown in the last column of the table. While the combined resummed calculation necessarily shows a larger uncertainty band than in either dQCD or SCET alone, the width of the band remains smaller than in pure fixed order. The results $\sigma_{\mathrm{NNLO+NNLL+EW}}$ correspond to the best prediction for the $t\bar{t}H$ production process that can be obtained to date.

The two panels of figure 2 plot the total cross section, the impact of the different higher-

---

[8]We remind the reader that, as discussed in Sec. 2.2, the soft scale is fixed to $\mu_s = Q/\bar{N}$ in the SCET approach.

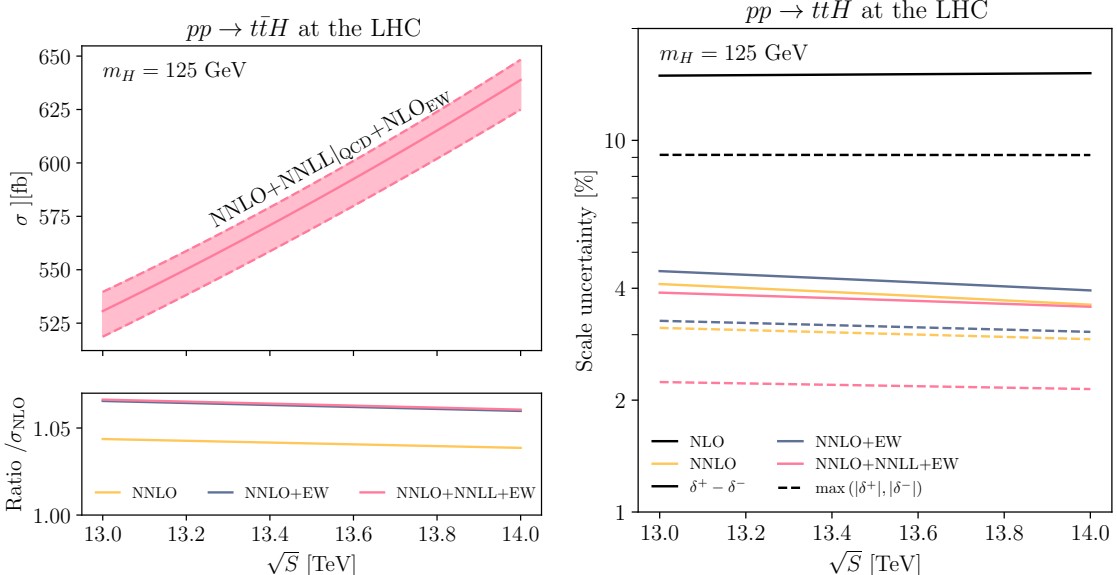

**Figure 2**: Left: the total cross section for $t\bar{t}H$, $\sigma_{\mathrm{NNLO+NNLL+EW}}$, plotted as a function of the collider energy $\sqrt{S}$ for $m_H = 125$ GeV. The inset shows the relative impact of the different contributions with respect to $\sigma_{\mathrm{NLO}}$. Right: scale uncertainties for the cross section, computed at different accuracies. Solid lines display the total width of the scale-uncertainty band, while dashed lines the maximum variation with respect to the central prediction.

order contributions and the residual scale uncertainties as a function of the collider energy, for the Higgs mass value $m_H = 125$ GeV. They give a visual summary of the discussion carried out so far: the left panel shows the absolute cross section and the impact of the different contributions, while the right panel shows the size of theoretical uncertainties. In this case, solid lines represent the total width of the scale-uncertainty band, while dashed lines stand for the maximum between the upper and lower scale variation.

In the following section, we will discuss the various sources of theoretical errors affecting these numbers, on top of the already mentioned scale variations.

## 3.3 Residual theoretical errors

In order to provide reliable predictions for the $t\bar{t}H$ cross section, a thorough estimate of all sources of theoretical errors is mandatory. In the previous section, we have already discussed the impact of missing higher orders in QCD, estimated via scale variations. Their smallness renders the assessment of the other possible sources of theoretical uncertainties even more crucial. We list and quantify relevant sources of theoretical uncertainties in the following. All quoted uncertainties have a negligible dependence on the specific Higgs mass and collider energies (when varied across the range of values considered in this work).

- *PDF and $\alpha_s$ uncertainties:* uncertainties due to partonic distributions and the value of the strong coupling are estimated following the PDF4LHC prescription. PDF uncertainties reflect the quality and consistency of the data employed for the fit. They amount to

$$\Delta_{\mathrm{PDF}} = 2.2\% \,. \tag{3.8}$$

  For what concerns the photon density, its minor impact together with the very precise determination stemming from the LuxQED method renders its uncertainty negligible. As far as $\alpha_s$ is concerned, again following the PDF4LHC recommendation, we quote uncertainties obtained from varying $\alpha_s(m_Z)$ by an amount of 0.001 with respect to the default value $\alpha_s(m_Z) = 0.118$. This variation is performed both in the PDFs and in the short-distance cross section. The uncertainty on the total cross section is

$$\Delta_{\alpha_s} = 1.7\% \frac{\delta\alpha_s}{0.001} \,. \tag{3.9}$$

- *Errors due to the approximation of the double-virtual contribution:* as discussed in section 2.1, in the NNLO calculation the two-loop amplitudes are estimated via two approximations that are ultimately combined through a weighted average. A corresponding systematic error is assigned by means of a conservative procedure that takes into account several sources of ambiguities, as detailed in Ref. [44]. The final error on the NNLO cross section turns out to amount to

$$\Delta_{\mathrm{virt}} = 0.9\% \,, \tag{3.10}$$

  and it is widely independent of the parameters in the range of the scan over collider energies $\sqrt{S}$ and Higgs boson masses $m_H$.

- *Numerical uncertainties:* the fixed-order result has been obtained within the $q_T$-subtraction formalism. In practice, the computation is performed [88] by introducing a technical cut-off $r_{\mathrm{cut}} = q_T^{\mathrm{cut}}/Q$ on the dimensionless variable $q_T/Q$, where $q_T$ $(Q)$ is the transverse momentum (invariant mass) of the $t\bar{t}H$ system. The final result, which corresponds to the limit $r_{\mathrm{cut}} \to 0$, is extracted by simultaneously computing the cross section at fixed values of $r_{\mathrm{cut}}$ and then performing an extrapolation to that limit. The error associated with this procedure, which combines statistical and extrapolation uncertainties, varies slightly between the setups, but is always small ($\mathcal{O}(0.3\%)$) compared to the other sources of theoretical uncertainties.

- *Ambiguities in the resummation approach:* as we have discussed in section 3.2, in particular when discussing table 2 the two different resummation procedures employed have a marginal effect on the total cross section, while they reduce the scale uncertainty band. Specifically, the effect on the cross section is at most at the 0.1% level. Moreover, the scale uncertainty band takes into account all scale variations from the two methods, and has thus to be regarded as very conservative

- *Uncertainties related to the top-quark mass value:* we estimate the dependence of the cross section on the top-quark mass by reporting how the cross section varies when $m_t$

is changed by 1 GeV with respect to the reference value. The top-quark mass enters both the kinematics part of the cross section (the dominant impact is from the phase space) and the top-quark Yukawa coupling, and the two effects have opposite sign. Remarkably, they tend to cancel almost exactly at the energies we considered, so the top-mass dependence can be neglected for the SM predictions, when the relation $y_t = \frac{\sqrt{2} m_t}{v}$ is enforced.

Still, it is also worth considering the case when the top-quark mass and the top-quark Yukawa coupling are varied independently. In this case, we obtain

$$\Delta_{y_t} = 1.1\% \frac{v}{\sqrt{2}} \frac{\delta y_t}{1\,\mathrm{GeV}}\,, \qquad \Delta_{m_t} = -1.1\% \frac{\delta m_t}{1\,\mathrm{GeV}}\,. \qquad (3.11)$$

The opposite sign of $\Delta_{y_t}$ and $\Delta_{m_t}$ reflects the opposite slope of the cross section when $y_t$ or $m_t$ are increased.

We note that, strictly speaking, this procedure is inconsistent when EW corrections are included. The effect of $y_t$ and $m_t$ variations, therefore, is assessed by neglecting them. However, given their rather small impact, EW corrections do not alter significantly the dependence of the cross section on $y_t$.

- *Uncertainties related to the top-quark mass renormalisation scheme*: As mentioned in Sec. 3.1, we worked with the top-quark mass and Yukawa couplings renormalised in the on-shell scheme. An alternative scheme to employ is $\overline{\mathrm{MS}}$, which is usually the reference scheme for the case of lighter heavy quarks (e.g. the bottom), since it resums to all orders logarithms involving the ratio $\frac{m^2}{\mu_R^2}$, $m$ being the heavy-quark mass (see e.g. Refs. [124–126]). In the case of $t\bar{t}H$, due to the large top-quark mass, these effects are expected to be negligible. Furthermore, any effect related to the renormalisation scheme must be higher-order with respect to the perturbative order at which predictions are computed. Results presented in Ref. [127], where the $t\bar{t}H$ cross section is computed at NLO in both schemes, help us giving a more quantitative statement. If we consider the total rate at NLO, changing the scheme amounts to a 1% effect. At NNLO such an effect is expected to be further reduced, hence negligible. We stress, however, that if differential observables are considered, larger effects may appear.

- *Uncertainties due to missing higher-order EW corrections:* since in the $G_\mu$ scheme the EW coupling is kept fixed, the relative effect of scale variations for EW corrections is identical to the LO contribution, and it does not cover missing higher orders in $\alpha$. Here we provide an argument to estimate NNLO$_2$, the first contribution where EW effects enter at NNLO, and which corresponds to $O(\alpha \alpha_s)$ corrections to $\sigma_{\mathrm{LO}}$. In order to have a rough estimate of possible effects at this order, the typical procedure is to study the difference between additive and multiplicative combinations of NLO QCD and NLO EW corrections. In a multiplicative combination, an extra term of $O(\alpha \alpha_s)$ appears, which improves the scale dependence of NLO EW corrections. In our case, we consider the NLO QCD and NLO EW $K$ factors, whose numerical value

is the following:

$$K_{\mathrm{NLOQCD}} \equiv \frac{\sigma_{\mathrm{NLO}}}{\sigma_{\mathrm{LO}}} = 1.26 \,, \qquad K_{\mathrm{NLOEW}} \equiv \frac{\sigma_{\mathrm{LO+EW}}}{\sigma_{\mathrm{LO}}} = 1.02 \,. \qquad (3.12)$$

The additive and multiplicative combinations are defined by

$$K_{\mathrm{NLOQCD+EW}} \equiv K_{\mathrm{NLOQCD}} + K_{\mathrm{NLOEW}} - 1 = 1.28 \,, \qquad (3.13)$$
$$K_{\mathrm{NLOQCD\times EW}} \equiv K_{\mathrm{NLOQCD}} \times K_{\mathrm{NLOEW}} = K_{\mathrm{NLOQCD+EW}} + 0.005 \,.$$

Where the extra 0.005 (0.5%) precisely corresponds to the extra $O(\alpha\alpha_s)$ term. This term gives a rough estimate of the $\mathrm{NNLO_2}$ contribution. Considering that the overall impact of such an uncertainty is further diluted by the large NLO QCD corrections, we can conclude that missing higher-order EW contributions will amount at most to few per mille of the final prediction, and therefore can be considered as a subleading source of uncertainty.

To clarify the previous discussion and show its application to a practical case, we report our state-of-the-art prediction, equipped with the dominant sources of uncertainties, for the $t\bar{t}H$ total cross section in the SM at $\sqrt{S} = 13.6\,\mathrm{TeV}$ and $m_H = 125.09\,\mathrm{GeV}$:

$$\sigma_{\mathrm{NNLO+NNLL+EW}}^{\sqrt{S}=13.6\,\mathrm{TeV},\, m_H=125.09\,\mathrm{GeV}} = 592.1\,\mathrm{fb} \underbrace{{}^{+1.5\%}_{-2.2\%}}_{\Delta_\mu} \underbrace{\pm 2.2\%}_{\Delta_{\mathrm{PDF}}} \underbrace{\pm 1.7\%}_{\Delta_{\alpha_s}} \underbrace{\pm 0.9\%}_{\Delta_{\mathrm{virt}}}, \qquad (3.14)$$

where we *assumed* the parametric uncertainty on $\alpha_s$, $\delta\alpha_s = 0.001$.

## 4 Conclusions

The $t\bar{t}H$ production process is a sensitive probe of the top-quark Yukawa coupling and its cross section represents a key observable in LHC physics. In this paper, we have presented a state-of-the-art computation of this observable within the SM. This computation combines the recently obtained NNLO QCD corrections with NNLL soft-gluon resummation and complete-NLO corrections into a full NNLO+NNLL+EW prediction.

The starting point of our calculation is the recent NNLO QCD computation of Ref. [44]. The missing two-loop amplitudes are derived therein by using two independent approximations that are ultimately combined to obtain an estimate of the finite part of the two-loop virtual contribution and its uncertainty. All the remaining ingredients of the calculation are evaluated exactly. The ensuing NNLO QCD result is combined with soft-gluon resummation up to NNLL accuracy.

From the perspective of perturbative QCD, a particularly interesting outcome of our work is the first-ever comparison of SCET and dQCD-based soft-gluon resummations at NNLO+NNLL order, for a process involving the non-trivial colour structure characteristic of four coloured partons in the Born level amplitude. Although the two methods share a common starting point, namely the factorisation of the partonic cross section in the soft gluon emission limit, the implementation of renormalisation-group improved perturbation

theory underlying the resummations differs, so that the SCET formulas contain some corrections that are considered N³LL and higher in dQCD and vice versa. In spite of these systematic differences between the two frameworks, the numerical results agree remarkably well at NNLO+NNLL in QCD, as clearly seen in the left-hand panel of figure 1. In both cases the resummation stabilizes scale uncertainties compared to NNLO alone, especially when considered across a wide range of (parametrically) different scales, as is apparent from the right-hand panel of the same figure. We have thus taken a conservative approach to residual resummation errors, taking into account the systematic differences between dQCD and SCET in addition to scale variations.

The NNLO+NNLL QCD results are eventually combined with the complete-NLO corrections, whose effect, although small, must be included at this level of precision. From a purely phenomenological perspective, our main results can be found in table 2, which shows NNLO+NNLL+EW predictions as a function of the LHC collider energy and Higgs mass, including an estimate of uncertainties from even higher-order QCD corrections. Remarkably, these uncertainty estimates are at the ±1-2 percent level. Other sources of theoretical uncertainty have been discussed and quantified in section 3.3 – those related to PDFs and $\alpha_s$ are currently the dominant ones, followed by those stemming from the approximation of two loop amplitudes.

## 5 Citation policy

The present work consists of contributions from different collaborations, whose work needs to be acknowledged. If the results are employed for scientific publications, together with this work, one should cite Refs. [38–40, 43, 44, 48–55]. The relevant BIBTEX keys are:

```
Frixione:2014qaa,Frixione:2015zaa,Kulesza:2015vda,Broggio:2015lya,Broggio:2016lfj,
Kulesza:2017ukk,Frederix:2018nkq,Kulesza:2018tqz,Broggio:2019ewu,Kulesza:2020nfh,
Catani:2022mfv,Devoto:2024nhl
```

## Acknowledgements

This work has been carried out within the LHC Higgs Working Group, as a contribution to the Yellow Report 5. We thank the members of the working group for various discussions on this topic, and for pushing us to pursue the work presented here. In particular, MW and MZ are specially grateful to Judit Katzy, Davide Valsecchi, Josh McFayden and Sergio Sanchez Cruz for their work as experimental conveners within the $t\bar{t}H$ subgroup.

The work of S.D. has been funded by the European Union (ERC, MultiScaleAmp, Grant Agreement No. 101078449). Views and opinions expressed are however those of the author(s) only and do not necessarily reflect those of the European Union or the European Research Council Executive Agency. Neither the European Union nor the granting authority can be held responsible for them. R.F. acknowledges the support by the Swedish Research Council under contract number 2020-04423. A.K. acknowledges the support by the German Research Foundation (DFG) under grants KU 3103/1 and KU 3103/2. T.S.

kindly acknowledges the support of the Polish National Science Center (NCN) grant No. 2021/43/D/ST2/03375. The work of M.W. was supported by the German Research Foundation (DFG) under grant 396021762 - TRR 257: *Particle Physics Phenomenology after the Higgs Discovery*. D.P and M.Z. acknowledge financial support by the Italian Ministry of University and Research (MUR) through the PRIN2022 Grant 2022EZ3S3F, funded by the European Union – NextGenerationEU.

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
