# Peer review of "State-of-the-art cross sections for ttH: NNLO predictions matched with NNLL resummation and EW corrections"

_SciPost Physics Community Reports_

## Round 1 · Referee Report · Anonymous (Referee 1) · 2025-7-1

Strengths

1- It is a thorough combination of the up-to-date predictions for the ttH production process. 2- Uncertainties are carefully explained and taken into account. 3- It provides clear recommendations for experimental analyses.

Weaknesses

1- It is based on a single NNLO calculation which still applies an approximation.

Report

The paper is a community effort that collects the current state-of-the art theoretical predictions for the cross section of associated production of a top-antitop pair with a Standard Model Higgs boson at the LHC (ttH for short). In my opinion, it satisfies the publication criteria of this journal.

The analysis includes the fixed-order NNLO corrections plus effects from the resummation of logarithmic terms due to soft-gluon emission. Also included are all lower-order effects, which the paper refers to as "complete-NLO" (properly defined in Eq.(2.24)). The fixed-order NNLO corrections are taken from Ref.[44] which still uses an approximation for the actual two-loop contribution, whose uncertainty is estimated and accounted for in the final prediction. Concerning the NNLL resummation, the authors pursue two
conceptually very different approaches, finding only small numerical differences in their numerical effects though.

The central part of the paper is section 3, where numerical recommendations for the inclusive total cross section at the LHC are provided, including a thorough breakdown of all theoretical uncertainties.

The paper is the result of a collaboration of a relatively large group of people (16 authors) which I believe are all part of the LHC Higgs Working Group. Such efforts are very desirable in order to clearly identify the source of any possible differences in different theoretical predictions and provide a clear recommendation to the experimental community which takes any systematic ambiguities into account. I therefore recommend the paper for publication without changes.

Recommendation

Publish (meets expectations and criteria for this Journal)

---

## Editorial Decision

resubmitted